# Introduced Herbivores Threaten the Conservation Genetics of Two Critically Endangered Single-Island Endemics, *Crambe sventenii* and *Pleudia herbanica*

**DOI:** 10.3390/plants13182573

**Published:** 2024-09-13

**Authors:** Priscila Rodríguez-Rodríguez, Sonia Sarmiento Cabello, Stephan Scholz, Leticia Curbelo, Pedro A. Sosa

**Affiliations:** 1Instituto Universitario de Estudios Ambientales y Recursos Naturales (IUNAT), Universidad de Las Palmas de Gran Canaria, Campus Universitario de Tafira, 35017 Las Palmas de Gran Canaria, Canary Islands, Spain; sonia.sarmiento@ulpgc.es (S.S.C.); leticia.curbelo@ulpgc.es (L.C.); pedro.sosa@ulpgc.es (P.A.S.); 2Jardín Botánico de Oasis Wildlife Fuerteventura, Carretera General de Jandía s/n, La Lajita, 35627 Pájara, Canary Islands, Spain; marmulano@gmail.com

**Keywords:** Brassicaceae, the Canary Islands, conservation genetics, Lamiaceae, microsatellites, species distribution modeling

## Abstract

*Crambe sventenii* Pett. ex Bramwell & Sunding and *Pleudia herbanica* (A.Santos & M.Fernández) M.Will, N.Schmalz & Class.-Bockh. are two single-island endemic species from Fuerteventura (Canary Islands), inhabiting the same areas and similar habitats. They are under the “Critically Endangered” category due to historical herbivore pressure, mainly goats, leading to habitat fragmentation and poor population recruitment. The main aim of our study was to provide insights into the conservation genetics and habitat suitability of these two species. For this purpose, we sampled all known populations on the island and developed two new sets of microsatellite markers. Moreover, to assist restoration plans, we performed species distribution models to determine the most suitable areas for reintroduction. While *Crambe sventenii* is highly fragmented, with low genetic diversity indices in some populations, *Pleudia herbanica*’s genetic structure is quite homogeneous, grouped in three main regions, with signs of inbreeding and an overall low genetic diversity. Both species could present moderate to high levels of autogamy. Our findings can provide guidance to local governments regarding conservation actions to be implemented in the field, like the identification of propagule sources and new suitable areas for restoration.

## 1. Introduction

Insular ecosystems are a key source of biodiversity globally, representing 25% of vascular plant species [1]. Despite their significance, island ecosystems are also highly vulnerable to anthropogenic disturbances due to their isolation and small population sizes [2]. The Canarian Archipelago is located in the Macaronesian Biogeographic region, <100 km off the northwestern coast of Africa. It is home to around 1300 native vascular plant species, with 44% of these endemic, many of which occur only on a single island [2,3,4]. This makes the Canary Islands home to over 50% of all endemic plant species in Spain, despite being only 1.5% of the national territory. However, around 26% of the Canarian flora is threatened, leading to the islands possessing the highest concentration of endangered species per area in Spain [5].

One of the major threats to Canarian flora is the foraging by introduced mammalian herbivores, mainly feral goats and the European rabbit [4], both considered among the 100 most invasive species worldwide [6]. Indeed, grazing by introduced herbivores has been shown to cause severe damage to the endemic flora [7,8,9]. Endemic insular plants have evolved without the presence of herbivores, so they did not develop mechanisms against them [10,11]. It has been shown that endemic plant species are more often consumed by herbivores that are non-endemic [12], decreasing seedling and plant species richness [13].

Moreover, plant populations species that are palatable to goats usually remain undevoured in inaccessible places, like rocky slopes where goats cannot access [14]. This situation can lead to severe habitat fragmentation and isolation if the distance between the living remnants is greater than the dispersal capacity of the species. Consequently, habitat fragmentation will cause genetic drift with the subsequent inbreeding and fitness reduction in the progeny [15], or even shift the evolutionary mating systems in plants, with a trend toward selfing [16].

*Crambe sventenii* and *Pleudia herbanica* (Figure 1a and Figure 1b, respectively) are two endemic plant species from Fuerteventura (The Canary Islands, Spain) of special concern and are critically affected by introduced herbivore grazing [17,18]. Both species fall into the category of “Critically Endangered” by the IUCN [19,20] and the Canarian catalog of protected species [21] due to their isolated distribution, low number of individuals, and population decreases.

*Crambe sventenii* B. Petter. ex Bramwell and Sunding (1973), (Brassicaceae), is a chamaephyte of 50 to 70 cm, hermaphrodite with possible self-compatibility, entomophilous, and its fruit is dispersed by birds [18,22]. It belongs to the section *Dendrocrambe*, which includes 14 species, with single-island or multiple-island endemics to The Canary Islands [22]. It is distributed in six localities with a low number of individuals (<500).

*Pleudia herbanica* (A. Santos and M. Fernández) M. Will, N. Schmalz, and Classen-Bockhoff [23], (Lamiaceae), is a small long-lived shrub, insect-pollinated by Hymenoptera and Lepidoptera [23]. The seeds are parasitized by *Oxyaciura tibialis* (Diptera: Tephritidae), which together with the predation by herbivores, is reducing the regenerative capacity of this species [17]. In the most recent monitoring efforts, 1200 individuals were counted throughout 10 localities, although only 200 are reproductive specimens [24]. It is a resilient species, adapted to the xeric conditions of the habitat [23]. The related *Pleudia aegyptiaca* is a native species in the Canaries, but hybridization with *P. herbanica* has not been detected.

Both species reside in very similar habitats in the south-east and south-west of Fuerteventura, considered the most arid region in Europe. Accordingly, the mean annual precipitation is less than 100 mm/year and irregular, with a mean annual temperature of around 20 °C and intense and constant winds [25]. Due to their isolated distribution and small population size, we predict high levels of inbreeding and genetic diversity loss in these species.

The aims of this study were to (1) analyze the genetic diversity and structure of *Crambe sventenii* and *Pleudia herbanica* populations from Fuerteventura to characterize their conservation genetics status; (2) evaluate the consequences of prolonged herbivory over their conservation genetics; and (3) estimate the habitat suitability of both species to optimize future reintroduction or restoration plans.

## 2. Materials and Methods

### 2.1. Sampling and DNA Isolation

We collected samples from a representative number of individuals from every known population in the south of Fuerteventura (Appendix A, Figure 1d). In total, we sampled 93 individuals of *Crambe sventenii* (Figure 1a) from 6 sites and 234 individuals of *Pleudia herbanica* (Figure 1b) from 11 sites. We included some samples located in the Botanical Garden of Fuerteventura with admixed origins from various populations. According to the most recent censuses, we covered 19.82% and 19.61% of *C. sventenii* and *P. herbanica* population sizes, respectively. In 2021, after the sample collection for this project, new populations were found at a lower altitude [26]. The coordinates of these new populations were considered to perform the species distribution models in *P. herbanica*. Young, dried leaves were desiccated in silica gel for storage and GPS coordinates were annotated per populations and individuals when it was possible due to the difficult topography. Genomic DNA of *C. sventenii* was extracted using the Invisorb DNA Plant HTS 96 Kit (STRATEC Molecular, Berlin, Germany); however, in *P. herbanica*, we used a DNA extraction protocol based on Dellaporta et al.’s research [27].

### 2.2. Microsatellite Development

Microsatellite loci were selected from an Illumina paired-end shotgun library developed by the company AllGenetics (University of A Coruña, A Coruña, Spain) using two probe mixes of each species. We initially chose 50 primer pairs per species from this library and labeled them with the protocol described in [28]. The initial PCR tests were conducted individually with each primer pair in a 25 uL total volume with the PCR Master Mix (Thermo Fisher Scientific, Waltham, MA, USA). Twenty and fifteen primer pairs (*Crambe sventenii* and *Pleudia herbanica*, respectively) were amplified consistently with more than two alleles and were selected to complete the genotyping for all samples (Appendix A).

Those primer pairs were amplified in four multiplex reactions for *C. sventenii* and three multiplex reactions for *P. herbanica*. Multiplex reactions were performed in a 15 uL total volume using QIAGEN Multiplex Kit (QIAGEN, Hilden, Germany). PCR conditions of the singleplex and multiplex reactions were conducted following [28] and the manufacturer’s instructions. PCR products were detected on an ABI 3730 Genetic Analyzer and fragments were sized against the LIZ (500_250) size standard (Applied Biosystems, Foster City, CA, USA) and visualized using Genemapper 4.0 (Applied Biosystems, Foster City, CA, USA). We identified allele peak profiles at each locus and assigned a genotype to each individual.

### 2.3. Genetic Analysis

The following genetic diversity indices were calculated using software GenAlex 6.5 [29]: total number of alleles (NTA); mean number of alleles per locus (Na); number of private alleles (NPA); percentage of polymorphism (P); observed heterozygosity (Ho); and expected heterozygosity (He). Rarefied allelic richness (A), based on the minimum number of samples, was estimated with the software SPAGeDi 1.5 [30]. The selfing rate per population was also inferred in this SPAGeDi 1.5 with the method developed by [31]. In the case of *P. herbanica*, the genetic diversity indices are shown by population and summarized by the regions detected with the software STRUCTURE v2.3.4 [32]. The inbreeding coefficient (*F_IS_*) and the significance of the deviation were obtained from the Hardy–Weinberg equilibrium and GENEPOP 4.2 [33]. Null alleles were estimated with MICROCHECKER [34].

To assess if stepwise mutations affected the population genetic structure of our species, we used an allele size permutation test [35] in SPaGeDi 1.5. This test is based on the comparison of *R*_ST_ pairwise values, with the distribution of the *R*_ST_ (p *R*_ST_) obtained through 10,000 replicates of allele sizes among allelic states. A significant test (an observed *R*_ST_ > 95% of the p *R*_ST_) shows that stepwise mutations contributed to the genetic differentiation between populations, otherwise, genetic drift is a relevant factor in the genetic structure. The values of the coefficient of genetic differentiation (*F*_ST_) between pairs of populations were estimated in GENALEX 6.5. Moreover, to evaluate the presence of an isolation-by-distance pattern in both species, we plotted the pairwise linearized *F*_ST_ (expressed as *F*_ST_/(1 − *F*_ST_)) and the geographic distance divided by 6 or 10 distance classes in *C. sventenii* and *P. herbanica*, respectively, over the entire distribution range with the software SPAGeDi.

In order to estimate the genetic relationships among populations, we obtained Neighbor-joining (NJ) dendrograms based on the distance between shared alleles (DAS), with 100 bootstraps on locus. The individuals from the Botanical Garden were not included in this analysis. The matrices of genetic distance and dendrograms were obtained from the software POPULATIONS 1.2.8 [36] and the resulting trees were edited with FigTree 1.3.1 [37].

In addition, a principal coordinate analysis (PCoA)—using the covariance standardized method of pairwise codominant genotypic distances among individuals—was implemented with GENALEX version 6.5.

To estimate the population genetic structure of the populations, all the genotypes were screened with the Bayesian method implemented in STRUCTURE [32]. The chosen model assumed admixture among the populations and correlated allele frequencies. Five independent runs were conducted for each value of K (n° of localities + 1), and every analysis consisted of a burn-in initial period of 105 replicates and a run length of 106 replicates in the Markow chain (MCMC). The optimal number of K was estimated with the ΔK method [38] and visualized in STRUCTURE HARVESTER [39]. The results with the 5 independent runs for the best fit K were processed in CLUMPP 1.1.2 [40], which iterates among the 5 independent runs to estimate the coancestry levels.

Genetic similarities among populations were analyzed with a nested analysis of molecular variance (AMOVA) [41], implemented in Arlequin [42]. For *Pleudia herbanica*, analyses were carried out by estimating the variation in regions (west, south, and east), between populations within groups and within populations.

### 2.4. Species Distribution Modeling

#### 2.4.1. Niche Modeling

We developed models of habitat suitability for the study species to detect the most suitable areas for reintroduction and restoration of the populations. All the known locations were included, including the artificial population in the Fuerteventura Botanical Garden since the individuals survive and reproduce without further assistance. In the case of *Pleudia herbanica,* we also included new localities found after the development of the study, for a better calibration of the models. All analyses were performed in R version 4.4 (R Core Team, 2020, Vienna, Austria). The modeling procedure and projections were performed using biomod2 package [43].

#### 2.4.2. Predictor Selection

Thirty-five topo-climatic environmental raster layers retrieved from Patiño et al. [44] with a high resolution of 100 m were analyzed to determine which ones best explained the distribution of the two studied species. To achieve this, a principal component analysis (PCA) was performed to identify the variables most closely associated with the species’ distributions. To avoid multicollinearity, the correlation between variables was carefully considered.

#### 2.4.3. Modeling Procedure and Evaluation

For the purpose of investigating the link between species studied and its environment, four different algorithms were selected. The algorithms chosen were a traditional algorithm, the generalized linear model (GLM), its semi-parametric extension the generalized additive model (GAM) (which allows smoothing functions in the predictors), and two bagging and boosting approaches—the generalized boosting model (GBM) and random forest (RF). A repeated data-splitting procedure (cross-validation) was carried out to evaluate the models. As there was no independent dataset, the models were calibrated on 80% of the data (training dataset) and evaluated on the remaining 20% (validation set). The entire procedure was repeated ten times. For the performance evaluation of models, two performance criteria were used, true skill statistics (TSS) and ROC.

#### 2.4.4. Ensemble Modelling for Current Habitat Projections

To predict the current and future habitat suitability for *Crambe sventenii* and *Pleudia herbanica*, an ensemble modeling of four algorithms previously mentioned (GLM, GAM, GBM, and RF) was used. Only models with a TSS greater than 0.8 were retained for building the final ensemble.

## 3. Results

### 3.1. Genetic Assessment

#### 3.1.1. *Crambe sventenii*

The allele profiles obtained after DNA amplification indicated that *Crambe sventenii* is diploid as the profiles showed a maximum of two peaks. Out of the 20 pairs of primers (20 loci), there was a different degree of polymorphism in the different populations. The 20 markers were polymorphic in C-JBOT and C-SAL, while C-COL and C-CAR showed only 45% and 40% of polymorphic loci, respectively (Table 1). None of the loci showed evidence of null alleles, and therefore all loci were used in the subsequent analyses.

The genetic diversity indices estimated in *C. sventenii* showed heterogeneous results depending on the population (Table 1). The highest genetic diversity was found in the Fuerteventura Botanical Garden (C-JBOT), with He = 0.475. On the one hand, the natural population with the highest genetic diversity values was C-SAL with Ar = 2.71, He = 0.448, and eleven private alleles; on the other hand, C-CAR presented the lowest values of Ar = 1.37, He = 0.12, and one private allele, followed by C-COL with He = 0.125 and Ar = 1.44. There are exclusive alleles with high allele frequencies in all the populations, especially in C-SAL, C-VIG, and C-COL (Appendix A). Only two populations (C-COL and C-CAR) showed deviation from Hardy–Weinberg equilibrium, testing for heterozygote deficiency (*p*-value < 0.05). The selfing rate estimates varied from 1.80% (C-VIG) to 20.40% (C-COL).

The allele permutation test of the *R*_ST_ pairwise values performed in SPaGeDi to detect a phylogeographic signal was not significant (*p* = 0.406); therefore, the genetic differentiation between populations is better explained by genetic drift and or migration than stepwise mutations [35].

The pairwise *F*_ST_ values were high among all the comparisons, ranging from 0.279 (PEÑ-SAL) to 0.841 (COL-CAR) (Appendix A).

The results to infer the genetic relationship between populations, such as the Neighbor-joining dendrogram (Figure 2a) and the PCoA (Figure 2b), showed a clear differentiation of C-CAR with respect to the other populations. The first two axes of the PCoA accounted for a high proportion of the total variance (49.08%). Nonetheless, in the PCoA, C-VIG, and C-COL were also quite differentiated, while C-SAL, PEÑ, and OLI are related. All the individuals were grouped according to the geographic origins of the samples. The individuals from the Botanical Garden (C-JBOT) appear between the population “Montaña Cardón” (C-CAR) and the rest (Figure 2c).

Consequently, the Bayesian analysis implemented in the STRUCTURE software shows a clear assignation to all the individuals according to the geographic origin, considering both K = 5 and K = 7, which were the two clustering with the highest AK (Figure 2c and Appendix A). In this analysis, in contrast with the *F*_ST_ values, the populations OLI and PEÑ appear more related in the K = 5 (Figure 2c). The AMOVA results also showed a high differentiation among populations, with 59.7% of variation found among populations (Table 2).

The representation of *F*_ST_/1 − *F*_ST_ shows a clear pattern of isolation by distance, with higher differentiation values with increasing geographic distances (Appendix A).

#### 3.1.2. *Pleudia herbanica*

According to the allelic profiles, *Pleudia herbanica* is diploid. By considering 15 pairs of primers (15 loci), we found that while there was some variation in the genetic profiles among different populations, the variation was more similar within regions (Table 1). There were no loci with signs of null alleles across the studied populations.

The genetic diversity values varied depending on the region. The western region showed the lowest values (Ar = 2.08; He = 0.148), while the southern region presented the highest (Ar = 2.67; He = 0.391). Despite performing the analysis by region level, only one or two exclusive alleles per region were detected.

The selfing rate estimates were high for some localities but the results per region indicated that *Pleudia herbanica* presents between 22.20% (eastern region) and 45.00% (western region) autogamy. All regions showed highly significant deviation from the Hardy–Weinberg equilibrium for heterozygote deficits considering the *p*-value < 0.001.

The allele permutation test performed in SPaGeDi to detect a phylogeographic signal was not significant (*p* = 0.311), so the differentiation is better explained by genetic drift and or migration than stepwise mutations [35].

The highest *F*_ST_ values were found among the different regions, with the highest value between the west and the east, and the lowest value between the south and the east (Appendix A).

We found a pattern of isolation by distance, determined by the correlation between the mean values of the linearized *F*_ST_ and the geographic distance (Appendix A).

The PCoA in *Pleudia herbanica* does not show major differences within the population level but the differences between the three major groups (east, south, and west) can be appreciated (Figure 3b). Moreover, in the PCoA, the admixture of the Botanical Garden individuals is shown, although there is a closer relatedness with the eastern populations. In contrast with the STRUCTURE results for K = 2, the NJ dendrogram groups the southern populations (P-COL and P-VLAR) with the western region.

According to the Bayesian analysis performed in STRUCTURE, the most probable clustering was K = 2 and K = 3 (Figure 3c and Appendix A). Therefore, both clusters are represented in Figure 3c. K = 2 distinguishes between the eastern and the western populations, with the southern population appears to be a mixture of the two. In the K = 3 clustering, the three regions are well differentiated, just like in the representation of the PCoA. The P-JBOT individuals appear to be more related to the eastern and southern populations.

The hierarchical AMOVA, implemented to test the percentage of how much the genetic diversity is distributed across populations and regions, showed that most of the variation is retained within populations (82.56%); however, the variation between the regions established (west, south, and east) was greater than the variation within regions, being 13.37% and 4.07%, respectively (Table 2).

### 3.2. Habitat Suitability

For modeling the distribution of *Crambe sventenii*, the following five variables were selected: annual precipitation, precipitation of the warmest quarter, temperature seasonality, north-ness, and aspect (Appendix A). In the case of *Pleudia herbanica*, elevation, aspect, annual precipitation, and precipitation of the warmest quarter were identified as the best predictors of distribution (Appendix A). None of the selected variables showed significant correlation with each other (Appendix A).

The area with the highest habitat suitability is found in the southern part of Fuerteventura above the Jandía Peninsula but there are also new areas in the Jandía Peninsula and in the central part of the island (Figure 4).

## 4. Discussion

In this article, we have unraveled the conservation genetic status, the patterns of genetic structure, and the niche requirements in two highly endangered single-island endemics from Fuerteventura. Even though both species share a distribution range, similar habitat, and a high sensitivity to herbivory, they present differences from a population genetics perspective, which may have consequences in the decision-making process toward their recovery.

### 4.1. Crambe sventenii, Highly Fragmented with Low Population Sizes

The knowledge of the reproductive biology of the species is essential to interpret the factors influencing the genetic structure and variability. *Crambe sventenii* might present self-compatibility [18,45], which is reflected in the selfing rate obtained for each population. This index varies widely from one population to another, from 1.8% to approximately 20% (Table 1). It was not possible to calculate the C-CAR due to the low number of polymorphic loci detected in this population, a further indication that the degree of selfing in this population is possibly even higher. Although we have detected some degree of selfing, the species presents mostly sexual reproduction. Accordingly, *C. sventenii* populations are in Hardy–Weinberg equilibrium, with the exception, once again, of C-COL and C-CAR populations, which showed a heterozygote defect. This is consistent with the higher degree of inbreeding that these populations present. The consequences of habitat fragmentation and low population sizes in threatened species are well-known [16], driving the species to possible extinction. The degree of genetic diversity detected in each natural locality is strongly related with the population sizes, so those populations with more genetic diversity are those with larger population sizes, with the lowest values in C-COL and C-CAR. This reinforces the importance of increasing the population sizes, as maintaining effective population sizes determines the viability of the populations in the long term [46].

Genetic structure analysis, considering all populations, shows a high degree of differentiation between the populations, which is accentuated in the western populations C-COL and C-CAR. Gene flow between populations appears to be very low with high *F*_ST_ values (Appendix A), despite the relatively small geographical distances between populations (maximum 10.5 km between C-CAR and C-COL). These differentiation values coincide in general terms with the values obtained in the work carried out in the Biota-Genes project [47], in which they analyzed the same populations of *C. sventenii* using the variability of RAPD markers, enhancing the results obtained in our study. The presence of private alleles was also very high, also indicating the lack of recent gene flow between populations due to isolation, which is increasing in genetic drift and genetic diversity loss in the western populations. The values of genetic differentiation obtained between the natural populations of *C. sventenii* are among the highest detected in plant species in The Canary Islands [48,49], or even the related *C. tamadabensis* and *C. pritzelii* [50]. The entomophily and ornithocory of *C. sventenii* [18] arise in contrast with the high population fragmentation detected as the birds could contribute to a further distance dispersal [51]. Nonetheless, the effectiveness of the seed dispersal system in *C. sventenii* should be revised, as habitat fragmentation and degradation may cause shifts in the dispersal mechanisms of plant species [52]. The strong pattern of isolation by distance found supports the hypothesis that the species might have been connected in the past. The absence of *C. sventenii* specimens and a conserved natural vegetation in flatter areas due to herbivory and historical uses is strengthening the habitat fragmentation and population isolation. The modeled distribution indicates a greater extent of habitat suitability compared to the species’ current distribution, which could serve as a guideline for reintroduction and creation of a corridor between the existing populations.

### 4.2. Pleudia herbanica, Low Genetic Diversity and Self-Compatible

The genetic results of *Pleudia herbanica* indicate low genetic diversity overall. Specifically, this is supported by low polymorphism rates, with some populations presenting polymorphism rates below 30%. In fact, the allelic richness and expected heterozygosity values detected are among the lowest for Canarian taxa studied with microsatellites [48]. This low genetic diversity could be explained by the high selfing rates detected in all the populations. Considering the results at the region level, the species could present up to 45% of selfing. At present, there are no studies of reproductive biology or floral morphology in *P. herbanica*, but the background in the genus *Salvia*, to which it belonged [23], is of gynodioecy [53], self-compatibility [54], pseudocompatibility [55], and is a facultative xenogamous species [54]. Given the *Salvia* antecedents, the hypothesis of a ”mixed mating system” strategy should not be discarded. The varied values of selfing rate found per population in *P. herbanica* and *C. sventenii* could be explained by the reduction in population size due to herbivory, diminishing the outcrossing between unrelated individuals. This pattern was also detected in the autogamous *Viola cheiranthifolia* [7]. Moreover, especially in *P. herbanica*, the effects of the selfing rate are shown in the Hardy–Weinberg equilibrium, with signs of inbreeding in almost all populations.

The low genetic differentiation between populations within the three defined regions (west, south, and east) is reflected in all the genetic structure results and the low presence of private alleles. Nonetheless, just like in *C. sventenii*, the results indicate that the principal factor influencing the genetic structure is the geographic isolation, which in this case, would be the isolation of the three regions.

Given that *P. herbanica* is a long-lived perennial species with limited dispersal capacity, it primarily establishes itself on rocky outcrops and vertical walls, which are the only accessible habitats where the absence of herbivores allows its growth. As there are individuals that remain in the population for many years (>50 years according to Scholz, personal comment), it is very likely that they are located very close to each other in the population, increasing the degree of inbreeding, enhanced by the reproductive self-compatibility. This would lead to an increase in genetic closeness between individuals, an increase in the number of individuals with related genotypes, and a reduction in the genetic diversity of populations. High values of genetic differentiation between regions, influenced by the geographical distance and small intra-regional differences with low levels of genetic diversity, are consistent with the inferred scenario. Long-lived species are generally more capable of mitigating the effect of genetic erosion due to habitat fragmentation and small population sizes for a longer period [56]. Consequently, the negative genetic outcomes—such as higher inbreeding or fitness reduction—for *P. herbanica* are likely to manifest in the future.

### 4.3. Botanical Garden Reservoir

The role of the Botanical Gardens is changing, with an increasing implication in plant conservation [57] through living plant collections and seed banks to ensure reliable sources for recovery plans. Living collections are crucial for species that cannot regenerate well in the wild or the seeds do not store well, in which case, the collection may store as much genetic diversity as possible from the original population [58].

The Botanical Garden of Oasis Wildlife, Fuerteventura, constitutes a living reservoir of *C. sventenii* and *P. herbanica*, harboring part of the genetic pool of both species in a location free of goats and anthropogenic disturbances. In *C. sventenii*, those individuals constitute the population with the greatest genetic diversity of all those studied. These specimens originated from 12 individuals initially cultivated, which successfully established and originated hundreds of specimens living in the reservoir nowadays [47]. It is remarkable that with only 12 specimens, a significant increase occurred in the genetic diversity in a few generations. The artificial population is regenerating by seeds without further issues, so the population is stable, and the individuals are vigorous (Scholtz, personal observation). Therefore, we do not expect outbreeding depression issues in this population despite its admixture origin [59]. According to the population genetic structure results, these individuals are more related to C- VIG and C-CAR populations, which could be considered in restoration programs due to the singularity of C-CAR and the low number of natural specimens. Therefore, the locality in the Botanical Garden could be maintained as a genetic reservoir of *C. sventenii* in case it is needed for future reinforcements and study of the species.

For *P. herbanica*, Fuerteventura’s Botanical Garden also constitutes a remarkable genetic pool for the species as it harbors 34 alleles out of the 115 detected with our markers, and it was the population with the second highest heterozygosity. Although the genetic drift and fragmentation in the natural populations is not as evident as in *C. sventenii*, the botanical garden location could serve as a genetic source and a field laboratory to further study the biology of the species and meet the requirements to propagate it. In fact, this is the only Botanical Garden to host semi-natural individuals of this species as the staff place seeds in rock crevices every autumn, where they germinate when the conditions are favorable [47]. Moreover, this artificial source could serve as a living laboratory to further study the parasitism of *Oxyaciura tibialis* on *P. herbanica* seeds [17].

### 4.4. Implications for Conservation

All the in situ and ex situ conservation measures proposed in this study will not be fruitful unless an exhaustive control of herbivores is implemented in the natural and reintroduced populations, especially goats. As the complete eradication of local goats and rabbits is unlikely to occur in the short term and the natural populations are isolated and inaccessible, we propose alternative actions that could benefit the conservation genetics of *Crambe sventenii* and *Pleudia herbanica*. The reintroduction of new populations in areas with high habitat suitability that are known to be free of herbivores and accessible to the managers could serve as a short-term solution to enhance the genetic diversity and connectivity of those species.

It is recommended that the genetic source of the natural population should be preserved by incorporating seeds in the available seedbanks in The Canary Islands. Moreover, all the natural populations could benefit from increases in population sizes, especially the western populations, which presented lower genetic diversity values in both species due to the lower number of individuals, genetic drift, and isolation. Population reinforcements, if necessary, should consist of seeds of specimens from the same population in the case of *C. sventenii* and from the same region in the case of *P. herbanica*. Equally, further experimentation could be carried out at Fuerteventura’s Botanical Garden, with the aim of assessing the viability of populations of mixed origin.

In *C. sventenii*, as outlined in the recovery plan [60], it is necessary to conserve a higher number of populations than originally specified due to the high heterogeneity found and the number of private alleles across different populations. The existence of a greater number of exclusive alleles in C-SAL—combined with the facts that it is the population with the highest levels of genetic diversity, has the largest number of individuals, and has the most northerly locality—makes it one of the priority populations for the genetic conservation of the species on Fuerteventura. However, the strong genetic differentiation existing between all the natural populations establishes the need to preserve a greater number of populations and specimens, as established in the approved Recovery Plan. Given that geographic isolation is the main cause of the genetic structure of populations of *Crambe sventenii* in Fuerteventura, and fragmentation due to the effect of herbivores is the main cause of isolation and genetic differentiation in the species, we propose to establish new populations in new enclaves with a high habitat suitability (Figure 4), duplicating the current natural populations in a different geographic space, with similar environmental conditions, in the absence of herbivores; this would improve the functional connectivity between the populations but also maintain all of the genetic variation found.

For *P. herbanica*, we also propose to avoid artificial and deliberate mixing of seeds and individuals, but with the difference of restricting this procedure to the level of each of the three genetic regions detected (west, east, and south) separately. We found no genetic reasons to prevent the mixing of seeds and propagules between populations within each region, and possibly the exchange of genomes from individuals of the same genetic region could increase the genetic diversity of the species, reduce the detected rate of selfing, and possible inbreeding effects. In this way, we could establish the three regions as collection and genetic conservation areas for *P. herbanica* in Fuerteventura. Since each region hosts at least one population with numerous individuals, we propose that seed collection should concentrate on those populations with the highest levels of genetic diversity, i.e., P-RES in the west, P-VLAR in the south, and P-OLI in the east. Finally, the discovery of new populations with different niche requirements than the original areas has allowed us to refine the species distribution models [26], finding new possible areas for establishing populations away from herbivore pressure, especially goat herding.

## Figures and Tables

**Figure 1 plants-13-02573-f001:**
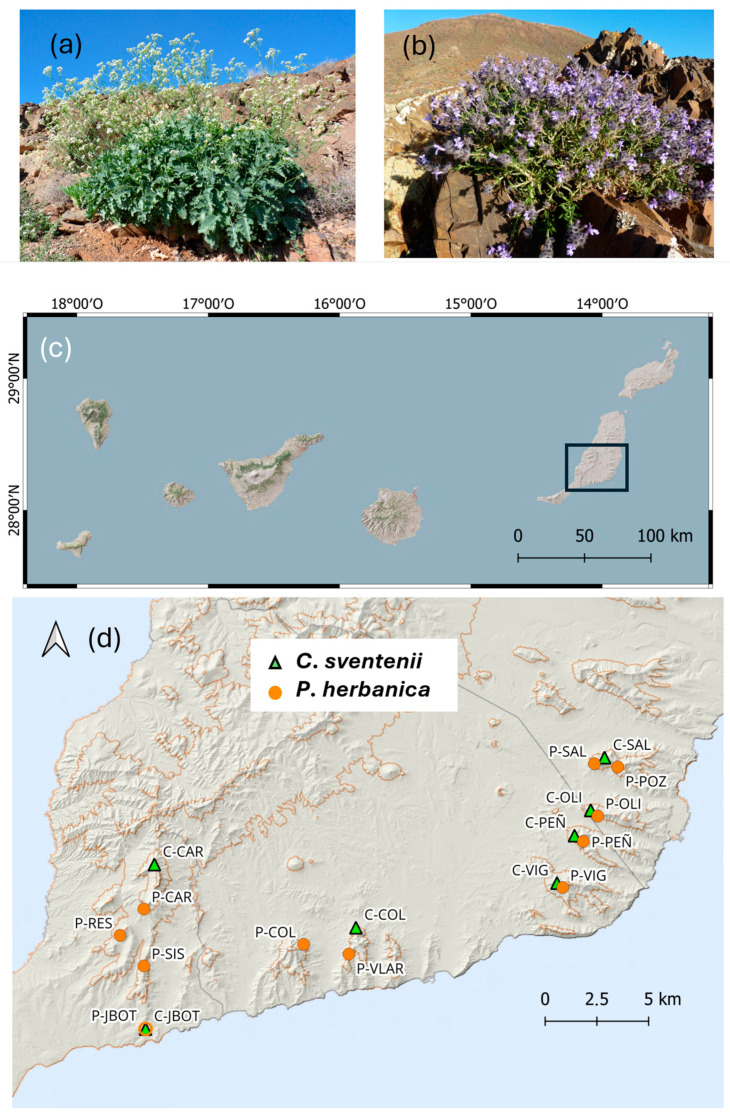
(**a**) *Crambe sventenii*. (**b**) *Pleudia herbanica*. Authorship: Stephan Scholz. (**c**) Map of The Canary Islands with a black square indicating the study area in Fuerteventura, Spain. (**d**) Location of the studied populations from *C. sventenii* (green triangle) and *P. herbanica* (orange circle).

**Figure 2 plants-13-02573-f002:**
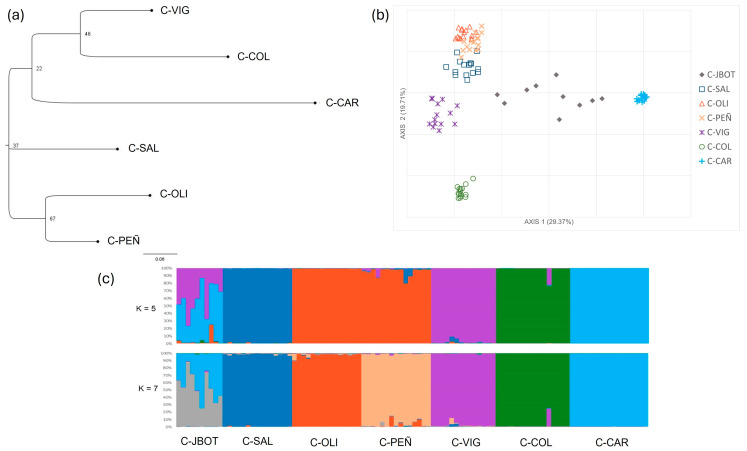
*Crambe sventenii* results. (**a**) Neighbor-joining dendrogram with populations, not including the Botanical Garden individuals. (**b**) Principal coordinate analysis (PCoA) for all *C. sventenii* sampled individuals. The first two axes explained 49.08% of the total variation. (**c**) Bar plots for the proportion of coancestry inferred from Bayesian cluster analysis implemented on STRUCTURE and CLUMPP, representing K = 5 and K = 7, following the highest ∆K (see Appendix A). Each color represents a different cluster.

**Figure 3 plants-13-02573-f003:**
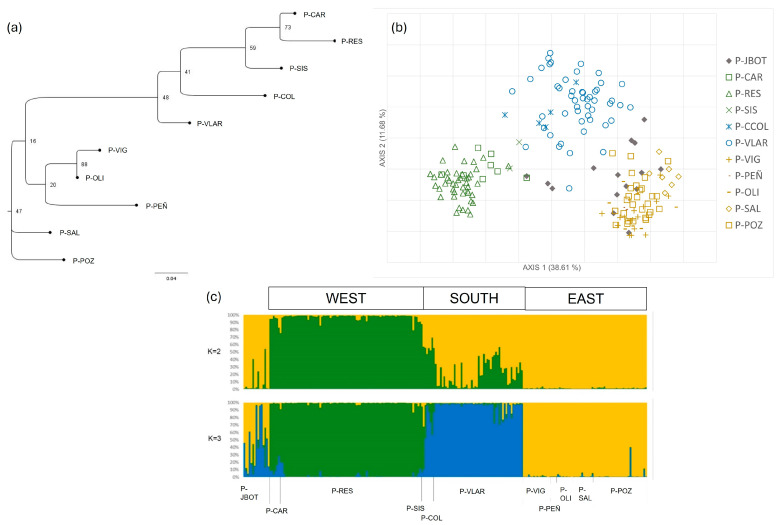
*Pleudia herbanica* results. (**a**) Neighbor-joining dendrogram with populations, not including the Botanical Garden individuals. (**b**) Principal coordinate analysis (PCoA) for all *P. herbanica* sampled individuals. The first two axes explained 50.29% of the total variation. (**c**) Bar plots for the proportion of coancestry inferred from Bayesian cluster analysis implemented on STRUCTURE and CLUMPP, representing K = 2 and K = 3, following the highest ∆K (see Appendix A). Each color represents a different cluster.

**Figure 4 plants-13-02573-f004:**
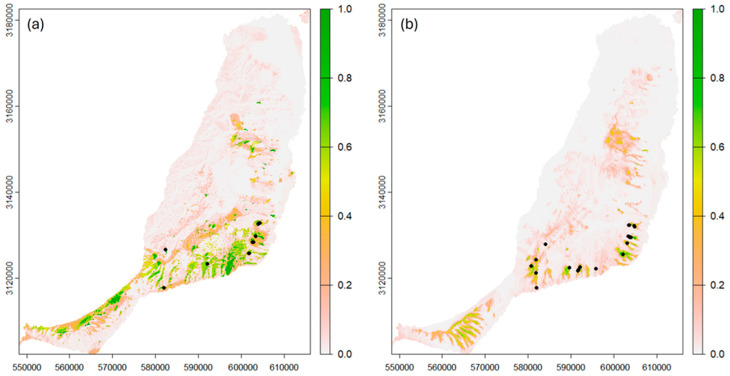
Output maps of the ensemble model of topoclimatic suitability of *Crambe sventenii* (**a**) and *Pleudia herbanica* (**b**). The black dots represent the coordinates used to indicate the presence of the populations.

**Table 1 plants-13-02573-t001:** Genetic diversity estimates of the *Crambe sventenii* and *Pleudia herbanica* populations.

Species	Region	Population	N	NTA	Na	NPA	Ar	P (%)	Sr (%)	Ho	He	*F_IS_*
*C. sventenii*		C-JBOT	10	48	2.40	1	2.36	100.00	2.60	0.474	0.475	0.002
		C-SAL	15	57	2.85	11	2.71	95.00	2.60	0.433	0.448	0.035
		C-OLI	15	41	2.05	2	1.95	80.00	19.40	0.293	0.301	0.025
		C-PEÑ	15	49	2.45	1	2.35	100.00	12.50	0.387	0.422	0.086
		C-VIG	14	45	2.25	4	2.17	75.00	1.80	0.321	0.328	0.020
		C-COL	16	30	1.50	4	1.44	45.00	20.40	0.119	0.125	0.054 *
		C-CAR	18	28	1.40	1	1.37	40.00	-	0.097	0.120	0.196 *
		**Total**	**103**	**94**	**2.13**	**24**	**-**	**76.40**	**9.88**	**0.304**	**0.305**	**-**
*P. herbanica*		P-JBOT	15	34	2.27	0	1.70	86.67	26.40	0.276	0.360	0.242 **
	West	P-CAR	9	20	1.33	0	1.24	33.33	-	0.081	0.134	0.405 *
		P-RES	80	28	1.87	1	1.24	66.67	37.70	0.099	0.129	0.234 **
		P-SIS	2	18	1.20	0	1.20	20.00	-	0.033	0.122	0.800
		**Total**	**91**	**66**	**2.13**	**1**	**2.08**	**80.00**	**45.00**	**0.096**	**0.148**	**0.356 ****
	South	P-COL	6	25	1.67	0	1.50	60.00	-	0.367	0.279	−0.358
		P-VLAR	50	39	2.60	2	1.76	93.33	32.50	0.290	0.392	0.262 **
		**Total**	**56**	**64**	**2.67**	**2**	**2.67**	**93.33**	**28.30**	**0.298**	**0.391**	**0.239 ****
	East	P-VIG	16	24	1.60	1	1.39	46.67	28.40	0.158	0.207	0.242 *
		P-PEÑ	4	18	1.20	0	1.16	20.00	-	0.133	0.090	−0.600
		P-OLI	8	26	1.73	0	1.46	53.33	49.20	0.116	0.238	0.532 **
		P-SAL	12	21	1.40	0	1.15	26.67	61.20	0.072	0.076	0.056
		P-POZ	30	26	1.73	0	1.42	60.00	-	0.188	0.220	0.146 **
		**Total**	**71**	**115**	**2.27**	**1**	**2.23**	**73.33**	**22.20**	**0.150**	**0.264**	**0.434 ****

N, number of analyzed samples; NTA, total number of alleles; Na, mean alleles per locus; NPA, number of private alleles; Ar, rarefied allelic richness; P, proportion of polymorphic loci; Sr, selfing rate; Ho, observed heterozygosity; He, expected heterozygosity; *F_IS_*, inbreeding coefficient. * *p* < 0.05; ** *p* < 0.01.

**Table 2 plants-13-02573-t002:** Analysis of the molecular variance (AMOVA) for *Pleudia herbanica* and *Crambe sventenii*. In *P. herbanica*, the hierarchy was established between the three regions considered (west, east, and south). ***: *p* < 0.001.

	Source of Variation	Degrees of Freedom	Sum of Squares	Variance Components	Percentage of Variation (%)	*F* Statistic
*P. herbanica*	Between regions	2	259.1	0.73	13.37	*F*_CT_ = 0.134 ***
	Between populations within regions	7	68.5	0.22	4.07	*F*_SC_ = 0.047 ***
	Within populations	428	1940.7	4.53	82.56	
	Total	437	2268.2	5.49		*F*_ST_ = 0.174 ***
*C. sventenii*	Between populations	5	663.4	4.20	59.7	
	Within populations	180	508.9	2.80	40.3	
	Total	185	1172.3	7.00		*F*_ST_ = 0.597 ***

## Data Availability

The original contributions presented in the study are included in the Appendix A, further inquiries can be directed to the corresponding authors.

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
