# Peer review of "Introduced Herbivores Threaten the Conservation Genetics of Two Critically Endangered Single-Island Endemics, Crambe sventenii and Pleudia herbanica"

_plants, 2024, doi:10.3390/plants13182573_

Round 1

Reviewer 1 Report

Comments and Suggestions for Authors

The authors present an interesting research dealing with the conservation genetics of endangered species, one of the most important aspects of conservation biology. Moreover, they put on the board the role of botanical gardens reservoir.

Focusing on two endemic species critically endangered of Fuerteventura (Crambe sventenii and Pleudia herbanica), the authors perform genetic analysis to clarify the genetic diversity and structure of the selected species. They also try to analyze the effects of long-term herbivory and the suitability of the habitat for future reintroduction or restoration. In addition, the authors discuss the role of botanical gardens as a reservoir of some threatened plant species.

The experimental design is appropriate, the analysis of data is adequate, and the tables and figures are necessary and well designed. The discussion and conclusions conclude the objectives of the study perfectly and the quality and extent of the bibliography, adequate.

In my opinion, the MS would be ready to become a paper for the journal Plants, but I'll suggest some minor details to improve it.

General aspects:

In the introduction you state that: "... makes the Canary Islands home to more than 50% of all 33 plant species endemic to Spain, despite being only 1.5% of the national territory. However, about 26% of the Canary flora is threatened, which means that the islands have the highest concentration of endangered species per area in Spain". Although it's true, the Canary Islands are in front of southern Morocco/Sahara occidental, several latitudinal degrees south of the continental country. I think that in order to inform readers, some sentence regarding this special location of the Canary Islands would be welcome.

Detailed aspects:

In the paragraph starting from line 85 regarding the meteorological characteristics of Fuerteventura, a quote is needed.

Well that's all, I enjoyed reading your research.

Author Response

We would like to thank the reviewer for taking the time to read our manuscript. His comments have encouraged us to improve the text. The changes are highlighted in yellow in the new version of the manuscript. Moreover, you can find below a detailed response to all the comments.

Reviewer 1:

Comments 1: In the introduction you state that: "... makes the Canary Islands home to more than 50% of all endemic plant species in Spain, despite being only 1.5% of the national territory. However, about 26% of the Canary flora is threatened, which means that the islands have the highest concentration of endangered species per area in Spain". Although it's true, the Canary Islands are in front of southern Morocco/Sahara occidental, several latitudinal degrees south of the continental country. I think that in order to inform readers, some sentence regarding this special location of the Canary Islands would be welcome.

Response 1: One sentence has been added to specifiy the geographic position of the Canarian archipelago in line 31: “The Canarian Archipelago is located in the Macaronesian Biogeographic region, <100 km off the northwestern coast of Africa”.

Comments 2: In the paragraph starting from line 85 regarding the meteorological characteristics of Fuerteventura, a quote is needed.

Response 2: We modified this sentence with the citation of a book chapter (Dorta, 2005).

Reviewer 2 Report

Comments and Suggestions for Authors

Abstract

Conclude on the aim (i.e. conservation genetics and habitat suitability of these two species)

Introduction

Unnecessary information abound and should be removed - see attached file.

P. herbanica  - clarify their conservation status, e.g. they fall into the category of “Critically Endangered” by the IUCN [25,26] and Canarian catalogue of protected species (BOC 4/2010).

Materials and Methods

We collected samples from a representative number of species (plants? populations?)

Did the target plants in the botanical Garden come from various populations or from just one population? Explain as this will have an impact on the genetic analysis.

Refer individually to Fig. 1a, 1b and 1c

Results

Mention Fig. 2c

Do not compare between the 2 target species in this section.

Individually mention Fig. 3a, 3b and 3c in this paragraph

No interpretation allowed in this section - just state what you found.

Discussion

Refer back to appropriate Figures and Tables

Unnecessary information abound and should be removed - see attached file.

Although we have detected some degree of selfing, the species presents mostly sexual reproduction, which makes most of the studied natural populations of C. sventenii in Hardy-Weinberg equilibrium, with the exception, once again, of C-COL and C-CAR, which showed a heterozygote defect, which is consistent with the higher degree of inbreeding that these populations seem to be suffering from. - confusing sentence because it is so long.

It is remarkable (why remarkable?) that the degree of genetic diversity detected in each natural locality is strongly related with population sizes ...

Gene flow between populations appears to be very low (WHY?) and, in some populations, practically non-existent (WHY?), despite the relatively small geographical distances between populations (maximum 10.5 km between C-CAR and C-COL). 

... generally coinciding (mean FST value = 0.566) with those detected in this work. So what does this mean? The presence of private alleles was also very high, also indicating the lack of recent gene flow between populations. Implications?

Additional issues are indicated (highlighted in yellow) in the accompanied reviewer's annotated manuscript.

Comments on the Quality of English Language

Minor language issues are indicated in the attached reviewer's annotated manuscript.

Author Response

We would like to thank the reviewer for taking the time to read our manuscript. His comments have encouraged us to improve the text. The changes are highlighted in yellow in the new version of the manuscript. Moreover, you can find below a detailed response to all the comments.

Reviewer 2:

Abstract

Comments 1: Conclude on the aim (i.e. conservation genetics and habitat suitability of these two species)

Response 1: The final sentence of the abstract has been modified to be more focuses on the applications of the study: “Our findings can provide guidance to local governments regarding conservation actions to be implemented in the field, like the identification of propagule sources and new suitable areas for restoration.

Introduction

Comments 2: Unnecessary information abound and should be removed - see attached file.

Response 2: Thank you for the suggestion, all the unnecessary text in this section has been removed as suggested.

Comments 3: P. herbanica  - clarify their conservation status, e.g. they fall into the category of “Critically Endangered” by the IUCN [25,26] and Canarian catalogue of protected species (BOC 4/2010).

Response 3: C: sventenii and P. herbanica have the same conservation status and the same categories in the IUCN red list and the Canarian Catalogue. It was stated in line 54-56 to avoid further repetition. The sentence was rephrased for clarity.

Materials and Methods

Comments 4: We collected samples from a representative number of species (plants? populations?)

Response 4: Whe meant “individuals”. Also, we have changed the sentences for a better English in lines 85-86: “We collected samples from a representative number of individuals from every known population in the Southwest of Fuerteventura”.

Comments 5: Did the target plants in the botanical Garden come from various populations or from just one population? Explain as this will have an impact on the genetic analysis.

Response 5: The plants came from various populations, so that information has been added in line 89

Comments 6: Refer individually to Fig. 1a, 1b and 1c

Response 6: We have changed all the figure mentions accordingly.

Results

Comments 7: You comment “Thus no significance” in the line 213 of the new version

Response 7: C-Col and C-CAR showed a significant deviation from HW, with a p-value <0.05. We have not made changes in this sentence as we think it is clear in the text and in table 1.

Comments 8 :Mention Fig. 2c

Response 8: The figure is now mentioned in the text.

Comments 9: Do not compare between the 2 target species in this section.

Response 9: All the comparisons between the two species have been removed from the Results section.

Comments 10: Individually mention Fig. 3a, 3b and 3c in this paragraph.

Response 10: The figures has been mentioned individually.

Comments 11: No interpretation allowed in this section - just state what you found.

Response 11: This sentence has been removed.

Comments 12: “Introduction,not a result”

Response 12: As suggested, we moved the sentence : “In the case of Pleudia herbanica, the genetic diversity indices are summarized by region….” to the Material and  Methods section.

Discussion

Comments 13: Refer back to appropriate Figures and Tables

Response 13: References to Figures and tables have been added.

Comments 14: Unnecessary information abound and should be removed - see attached file.

Response 14: We appreciate the suggestion of removing the first introductory paragraph of the introduction. However, we think that an introductory paragraph is useful for the reader. Even though we have deleted the first sentence to remove unnecessary content, in line 319.

In line 326, we have simplified the first sentence of this paragraph, which is “ The knowledge of the reproductive biology of the species is essential to interpret the factors influencing the genetic structure and variability”.

Comments 15: Although we have detected some degree of selfing, the species presents mostly sexual reproduction, which makes most of the studied natural populations of C. sventenii in Hardy-Weinberg equilibrium, with the exception, once again, of C-COL and C-CAR, which showed a heterozygote defect, which is consistent with the higher degree of inbreeding that these populations seem to be suffering from. - confusing sentence because it is so long.

Response 15: The sentence has been rephrased from line 332 to 336: “Although we have detected some degree of selfing, the species presents mostly sexual reproduction. Accordingly, C. sventenii populations are in Hardy-Weinberg equilibrium, with the ex-ception, once again, of C-COL and C-CAR populations, which showed a heterozygote defect. This is consistent with the higher degree of inbreeding that these populations present”

Comments 15: It is remarkable (why remarkable?) that the degree of genetic diversity detected in each natural locality is strongly related with population sizes ...

Response 15: We deleted the words “It is remarkable” in line 338, as it is an expected results in low population sizes

Comments 16: Gene flow between populations appears to be very low (WHY?) and, in some populations, practically non-existent (WHY?), despite the relatively small geographical distances between populations (maximum 10.5 km between C-CAR and C-COL). 

Response 16: The sentence has been rephrased : “Gene flow between populations appears to be very low   with high FST values, despite the relatively small geographical distances between populations (maximum 10.5 km between C-CAR and C-COL)”. Moreover, we hypothesize about this strong isolation from line 369, which we also modified to: “The strong pattern of isolation by distance found supports the hypothesis that the species might have been connected in the past. The absence of C. sventenii specimens and a conserved natural vegetation in flatter areas due to herbivory and historical uses is strengthening the habitat fragmentation and population isolation.

In addition, in this paragrapah we added a sentence that was in the results section before: “The modelled distribution indicate a greater extent of habitat suitability compared to the species current distribution, which could serve as a guideline for reintroduction and create a corridor between the existing populations”

Comments 17:... generally coinciding (mean FST value = 0.566) with those detected in this work. So what does this mean? The presence of private alleles was also very high, also indicating the lack of recent gene flow between populations. Implications?

Response 17: We have deleted the commented sentence, and added the following statement to the previous, to highlight the coincidence between the two studies, although they used different markers. “These differentiation values coincide in general terms with the values obtained in the work carried out in the Biota-Genes project [48], in which they analysed the same pop-ulations of C. sventenii using the variability of RAPD markers, enhancing the results obtained in our study”. Inline 353, we added the isolation as a cause for the low gene flow. Nevertheless, there is a higher explanation at the end of the paragraph.

Comments 18: In line 348, you wondered why Pleudia herbanica would go this route of reproduction (mix mating with selfing and outcrossing).

Response 18: This whole paragraph has been rephrased for a better understanding of the autogamy rates reasons. The changed were made from line 375 to 384: “Considering the results at the region level, the species could present up to 45 % of selfing. At present, there are no studies of reproductive biology or floral morphology in P. herbanica, but the background in the genus Salvia, to which it belonged [23], is of gy-nodioecy [53], self-compatibility [54], pseudocompatibility [55] and facultative xenog-amous species [54]. Given the Salvia antecedents, the hypothesis of a ‘mixed mating system’ strategy should not be discarded. The varied values of selfing rate found per population in P. herbanica and C. sventenii could be explained by the reduction in popu-lation size due to herbivory, diminishing the outcrossing between unrelated individu-als. This pattern was also detected in the autogamous Viola cheiranthifolia [7]. Moreover, especially in P. herbanica, the effects of the selfing rate are shown in the Hardy-Weinberg equilibrium, with signs of inbreeding in almost all populations.

Comments 19: You wondered who is Scholz in line 403.

Response 19: Stephan Scholz is the botanist leading the Fuerteventura Botanical Garden, and one of the authors of this article. He has a lifetime experience with the study species, with apreciations that are not published but are worth mentioning.

Comments 20: In “Consequently, the negative genetic consequences, for P. herbanica are likely to manifest in the future”. You commented “such as..” for the negative consequences.

Response 20: We have added the following: Consequently, the negative genetic consequences, like higher inbreeding or fitness reduction, for P. herbanica are likely to manifest in the future.

Comments 21: In 430, you commented “why not?” to the statement: “we do not expect outbreeding depression issues…”

Response 21: The reason is in the previous sentence: “The artificial population is regenerating by seeds without further issues, so the population is stable, and the individuals are vigorous”

Comments 22: In line 461: “Moreover, all the natural populations could benefit of an increase in the population size, especially the western populations, which presented lower genetic diversity values in both species”, you commented “Why do they have lower genetic diversity?”

Response 22: We have added a final statement in this part: Moreover, all the natural populations could benefit of an increase in the population size, especially the western populations, which presented lower genetic diversity values in both species, due to the lower number of individuals and genetic drift

Comments 23: Additional issues are indicated (highlighted in yellow) in the accompanied reviewer's annotated manuscript.

Response 23: All the issues highlighted in the manuscript has been addressed in the new version, and the changes can be seen in yellow. Moreover, additional responses to most of the comments has been included in this response letter.

Round 2

Reviewer 2 Report

Comments and Suggestions for Authors

The authors are thanked for the improvements.

Comments on the Quality of English Language

Minor language issues